# Inflammatory Factors: A Key Contributor to Stress-Induced Major Depressive Disorder

**DOI:** 10.3390/cells14090629

**Published:** 2025-04-23

**Authors:** Qian Liu, Baowen Nie, Xuemin Cui, Wang Wang, Dongxiao Duan

**Affiliations:** Department of Physiology and Neurobiology, School of Basic Medical Sciences, Zhengzhou University, Zhengzhou 450001, China; qian0825@gs.zzu.edu.cn (Q.L.); niebaowen@gs.zzu.edu.cn (B.N.); 18331066539@163.com (X.C.); zzuwwuzz@zzu.edu.cn (W.W.)

**Keywords:** cognitive dysfunction, inflammation, major depressive disorder, mental health, stress

## Abstract

Major depressive disorder (MDD) is a prevalent psychiatric disorder with a complex pathogenesis influenced by various factors. Recent research has highlighted a significant connection between psychological stress and MDD, with inflammation playing a central role in this relationship. Studies have demonstrated that peripheral immune changes in patients with MDD and in mouse models of social stress are closely linked to depressive symptoms. These findings suggest that targeting peripheral immune factors could represent a novel approach for treating stress-related neuropsychiatric disorders. Stress triggers a cascade of inflammatory responses, leading to disruptions in neurotransmitter metabolism and reduced synaptic plasticity. These changes exacerbate depression and contribute to cognitive decline. This study examines the bidirectional relationship between MDD and stress, focusing on the role of inflammation in this complex interplay. Recent studies have identified specific immune factors that are elevated in the serum of patients with MDD and stress-exposed mice, indicating a mechanism by which peripheral immune responses can affect central nervous system function and behavior. Furthermore, proteins, such as nuclear factor kappa-B (NF-κB), reportedly play a critical role in the regulation of stress hormones and are associated with depressive behaviors. Understanding these mechanisms is essential for advancing diagnostic, intervention, and treatment strategies for MDD.

## 1. Introduction

Major depressive disorder (MDD) is a severe mental illness affecting tens of millions of people worldwide. It often leads to diminished cognitive functioning, reduced quality of life, and significant burdens on patients, their families, and society. In China, where depression accounts for one-fifth of global cases, the prevalence of depression among youth is particularly alarming, with a detection rate of 24.6% and a major depressive disorder rate of 7.4%. These figures underscore the pressing need to address MDD not only as an individual mental health concern but also as a critical public health challenge that requires immediate attention and comprehensive intervention strategies. Early identification and effective treatment of MDD can significantly improve patients’ quality of life and social functioning.

Stress, defined as an individual’s response to external environmental stimuli, has been widely recognized as a key causative factor in the development of major depression, particularly when experienced over prolonged periods. Emerging research suggests that inflammatory responses may serve as a crucial link between stress and depression. Therefore, investigating the intricate relationship between stress, inflammatory processes, and depression—along with the underlying mechanisms of inflammation’s role—is of great academic and clinical significance. Such exploration has the potential to contribute to the development of innovative therapeutic approaches and strategies for managing MDD effectively.

## 2. Epidemiology of Depressive Disorder

Depressive disorder (DD) is a common mental illness that affects millions of people globally. According to the World Health Organization, approximately 350 million people suffer from depression, and it is projected that depression will become one of the most prevalent diseases worldwide by 2030 [1]. MDD, a severe form of depression, is characterized by significant pessimism, cognitive dysfunction, and various somatic symptoms. These symptoms include persistently low mood, anhedonia, lack of motivation, and diminished desire to engage in new activities. Sleep disturbances and increased suicide risk are also commonly observed. Additionally, physical symptoms, such as insomnia, appetite changes, and body aches, are prevalent, often leading to a substantial decrease in the patient’s quality of life [2].

Before 2015, psychiatric diagnoses were based primarily on subjective symptoms and observable signs. However, significant advancements in understanding depression have been made since then [3]. Despite these advancements, the methods used for diagnosing depression in clinical settings remain limited, largely due to the complex and multifactorial nature of the disease. This underscores the need for further research to deepen our understanding of its pathogenesis.

The prevalence of depression is influenced by multiple factors, each of which can elevate the risk of its onset [4]. Regular dietary habits can also increase or decrease the risk of depression [5]. Thus, studying the pathogenesis of depression requires a broad perspective, incorporating various social, environmental, and biological factors. Studies have shown that genetic factors, stress, and co-morbidities, among others, can contribute to the onset of MDD. Notably, existing research overwhelmingly emphasizes the relationship between stress and MDD [6,7,8]. Stress plays a critical role in depression and may increase the risk of suicide among affected patients [9].

At present, the main treatment methods of cognitive function are drug therapy, physical therapy, psychotherapy, and exercise therapy, and the main treatment methods of MDD. Many MDD patients have improved their mood after antidepressant treatment, but the improvement of cognitive function is not necessarily consistent. So far, it has been proved that two antidepressants, vortioxetine and duloxetine, can lead to direct, independent, and clinically relevant changes in cognitive dysfunction in MDD patients [10].

## 3. Impact of Stress on Depression

The impact of stress on depression can be understood through two main types of stress reactions: acute stress and chronic stress. Acute stress is typically an immediate response to an unexpected event and is characterized by rapid physiological changes, such as an increased heart rate and shortness of breath. These changes are primarily mediated by the sympathetic nervous system and are designed to help an individual respond swiftly to a perceived threat or challenge. In contrast, chronic stress represents a sustained state of strain that results from prolonged exposure to stressors. Chronic stress is linked to various health problems, including depression, anxiety, and cardiovascular disease. The physiological mechanisms of chronic stress involve alterations in the endocrine system, particularly an increase in cortisol secretion [11]. Chronically high levels of cortisol can suppress the immune system, which in turn negatively affects overall health. Therefore, understanding the different types of stress responses and their physiological mechanisms is essential for developing effective coping strategies and interventions.

### 3.1. Sex-Specific Effects on Depression During Stress

There are notable gender differences in biomarker levels, indicating that gender plays an objective role in the development and progression of MDD [12]. The prevalence of MDD is 2–3 times higher in female compared to male individuals. In male individuals, the prevalence of MDD remains relatively stable from adolescence to old age. However, in female individuals, the prevalence of MDD surpasses that of males from adolescence through menopause. The risk of depression in female individuals varies throughout their lifespan, with hormonal changes, such as those occurring postpartum, contributing to the onset of common postpartum depression [13]. Teenage girls are at a higher risk for depression, anxiety, self-harm, and suicidal behavior compared to teenage boys [14]. Furthermore, female patients with MDD tend to experience more severe symptoms than their male counterparts and are more likely to have co-morbid anxiety [15]. Chronic social stress, particularly from experiences such as bullying, is a significant risk factor for the development of mood disorders. Female rodents have been found to be more susceptible to unpredictable stressors, exhibiting anxiety- and depression-like behaviors as early as 6 days, while male rodents do not show these behaviors [16]. Studies have shown that comparing MDD patients under social stress tests, female MDD patients have higher levels of salivary cortisol than male MDD patients [4].

Research has indicated that the elderly, pregnant women, and children are more likely to experience MDD, which may be linked to genetic factors, physical health, and social environments [17]. Sex-specific genetic differences have been identified, with studies highlighting genes, such as *EIF4EBP2*, *CYP24A1*, *HSPA12B*, *RPS26*, and *ZNF729*, as being associated with MDD in both men and women (Figure 1A) [18].

### 3.2. Expression Changes in Stress-Related Inflammatory Factors in MDD

MDD is a complex, multifactorial disorder, and genetic factors play a role in its development. Twin studies suggest that MDD has a heritability of approximately 30–40% [19], making the investigation of genetic variations and gene expression crucial for understanding the pathogenesis and regulatory mechanisms of MDD. In families with a high prevalence of MDD, the disorder is more likely to manifest compared to members of the general population, possibly due to long-term exposure to stressful environments. Abnormal gene expression in these families may contribute to the onset of MDD by reducing the production of growth factors during brain development, triggering the disorder [20].

Research has shown that the DNA damage-inducible transcript 4 (DDIT4) gene is regulated by oxidative stress. In individuals with depression, DDIT4 is activated in the mouse prefrontal cortex (PFC) in response to stress. Analysis of postmortem brain tissue from individuals with MDD revealed that the expression of genes such as DDIT4, NTRK2, and WSB1 was upregulated, while UGT8, PRPS1, and DUSP6 expression was downregulated (Figure 1B) [21]. Jansen et al. found significant differences in the expression of genes associated with MDD compared to healthy controls, suggesting that MDD-related gene expression involves a complex network of genes (Figure 1C,D) [22].

Individuals with MDD often experience disruptions in their biological rhythms, particularly in the regulation of circadian rhythms. Clock genes, which control these rhythms, encode proteins that interact in feedback loops, regulating various physiological processes in alignment with the Earth’s 24 h cycle. The expression of these genes fluctuates throughout the day, influencing physiological functions, such as the sleep-wake cycle, hormone secretion, body temperature, and metabolic activity. Li et al. analyzed bioinformatics data from the Gene Set Enrichment Analysis, which included 21 healthy individuals and 24 patients with MDD. They found that the expression of several clock genes, including histone deacetylase 1, inhibitory factor 3, interleukin (IL)-3, and protein kinase AMP-activated catalytic subunit alpha 1, was significantly different in patients with MDD compared to healthy controls [23].

Malki et al. observed that cortical cells in the brain are enriched with genes associated with MDD, suggesting the presence of different genetic mechanisms underlying abnormal brain function. This emphasizes the importance of cortical gene expression in brain development. For instance, the expression of the SLA gene was upregulated, while the expression of recombinant human RCAN2 was downregulated, leading to cortical developmental abnormalities [24]. Additionally, Yang et al. identified butyrophilin subfamily 3 member A2 (BTN3A2) as a key gene with significant relevance to the pathogenesis of MDD [19].

### 3.3. Effects of Stress on the Nervous System

The development of MDD is associated with structural and functional changes in several brain regions, including the prefrontal cortex, hippocampus, amygdala, and ventral striatum. For example, abnormalities in the hippocampus contribute to clinical symptoms, such as reduced attention, memory deficits, and decreased motivation in patients with MDD [25]. When an individual is exposed to a stressor, the inferior optic thalamus activates the hypothalamic-pituitary-adrenal (HPA) axis by releasing corticotropin-releasing hormone. This hormone prompts the adrenal glands to secrete cortisol, which plays a critical role in regulating energy metabolism and immune responses and influencing psychological states. Cortisol enhances the excitability of the limbic system, which in turn promotes the release of glutamate and enhances synaptic plasticity through acute stress and elevated CRF levels (Figure 2) [26]. These responses ensure that individuals can effectively cope with challenges in their environment.

Dudek et al. demonstrated that chronic social stress alters the vascular ultrastructure and disrupts the blood–brain barrier (BBB) in the nucleus accumbens of male mice [27]. In addition, peripheral myeloid cells and pro-inflammatory factors can infiltrate the brain, contributing to BBB dysfunction. Chronic social stress downregulates the tight junction protein Cldn5. This disruption, coupled with stress-induced peripheral immune signaling, increases BBB permeability, which may lead to depressive-like behaviors [28]. As depression persists over time, hippocampal volume decreases, gene expression becomes altered, and neurogenesis decreases. Mahajan et al. found that 30 genes were downregulated in individuals with MDD compared to healthy controls. Several of these downregulated genes are closely related to inflammation, such as ISG15, IFI44L, IFI6, and NR4A1/Nur-77 (Table 1). Enrichment analysis revealed that signaling pathways associated with inflammation and neurogenesis, including the mitogen-activated protein kinase (MAPK) pathway, were highly expressed in the hippocampal dentate gyrus of patients with MDD [29].

## 4. Association of Stress-Mediated Inflammation with Depression

### 4.1. Relationship Between Stress and Inflammation

Inflammation is triggered by stress or abnormal bodily functions. Numerous studies have demonstrated that psychological and social factors, along with chronic stress, influence various biological processes in humans, such as inflammation and metabolic changes [34]. Chronic psychological stress also impacts disease progression, recurrence, and treatment outcomes and is associated with systemic inflammation, potentially accelerating the development of inflammation-related diseases [35]. During pregnancy, mothers experience a constant physiological response to external stressors, which are characterized by their ability to trigger inflammation. This genetic continuity may increase the risk of mental disorders in offspring. Furthermore, the mild inflammation induced by psychological stress is typically more persistent and has a greater impact on the offspring than the transient inflammation caused by acute infection or injury [36].

Inflammation is characterized by the infiltration of inflammatory cells and release of inflammatory mediators, representing a complex defense response in tissues to injurious factors. Activated inflammatory cells produce both pro-inflammatory and anti-inflammatory factors, and the balance between these factors helps the body regulate inflammation. Inflammation affects the nervous system through various pathways, including the release of pro-inflammatory factors, changes in neuronal plasticity, and alterations in neurotransmitter synthesis and metabolism. These inflammatory factors can initiate neuroinflammation by activating the immune system, thereby impacting brain structure and function.

Moreover, inflammation may exacerbate the pathological process of depression by affecting brain-derived neurotrophic factor (BDNF) levels, which in turn hinders neuronal growth and survival [37]. When inflammatory cells are activated, they generate various inflammatory mediators, including ILs, interferons, and growth factors. Elevated pro-inflammatory cytokines, such as IL-1β, IL-6, and IL-18, are frequently observed in patients with depression and may trigger neuroinflammatory processes along with peripheral inflammation [38]. Specifically, IL-18 has been linked to cognitive impairment in MDD. Fulgenzi et al. found that a deficiency in IL-18 could result in hippocampal damage, leading to depression-like behavioral changes. Additionally, research suggests that IL-18 knockout mice exhibit more severe and sustained stress-induced neuroinflammation and glucocorticoid signaling, both of which contribute to behavioral changes (Figure 3) [2].

### 4.2. Association of Stress with Signaling Pathways

Inflammation plays a crucial role in the development of depression. Research has shown a significant correlation between elevated inflammatory markers in the body and exacerbated depressive symptoms, particularly in individuals with chronic illnesses. Inflammatory factors affect brain function through various mechanisms, such as influencing neurotransmitter synthesis and metabolism, altering neuroplasticity, and impairing the neuronal environment. Several signaling pathways are involved in stress response, including the Ephrin-B2/EphB4 signaling pathway, Akt, mTOR, nuclear factor kappa-B (NF-κB), PI3K/AKT, HMGB1/STAT3/p65, SAPK/JNK, and MAPK family signaling pathways. Among these, the NF-κB signaling pathway is particularly significant and has been extensively studied. It is one of the most important and well-established pathways, and thus, this review will focus on its role in depression.

#### 4.2.1. Relationship Between NF-κB and Major Depressive Disorder

NF-κB is a transcription factor that plays a key role in a variety of biological processes, including inflammatory responses, cell survival, and immune regulation. In recent years, there has been increasing interest in the role of NF-κB in major depressive disorder. The activation of NF-κB is closely associated with the expression of various inflammatory factors, which contribute significantly to the pathogenesis of depression. Elevated serum levels of inflammatory markers have been observed in patients with depression, which may be linked to the activation of the NF-κB signaling pathway [39]. Moreover, excessive activation of NF-κB can lead to neuronal damage and apoptosis, which may exacerbate depressive symptoms [40]. Shoji et al. demonstrated that NF-κB influences neuronal survival and growth by regulating the expression of BDNF, a key molecule involved in neuroplasticity. In patients with depression, BDNF levels are reduced, but suppressing NF-κB can restore BDNF expression, thereby alleviating depressive symptoms [41]. This suggests that NF-κB may serve as a potential therapeutic target for major depressive disorder.

#### 4.2.2. NF-κB Pathway

The MAPK signaling pathway is one of the most critical cellular pathways involved in the stress response. The activation of MAPK is mediated by several pathways, including the PAK pathway, GCK pathway, and NF-κB-inducible kinase (NIK) pathway. Within the GCK pathway, cytokine tumor necrosis factor (TNF)-α activates JNK and the transcription factor NF-κB through MEKK1, which subsequently promotes the expression of inflammatory factors. Additionally, NF-κB can feedback-regulate the activity of the MAPK signaling pathway, forming a complex regulatory network. This interaction is essential in the cellular response to external stimuli, inflammatory reactions, and disease development.

NF-κB is a family of transcription factors comprising p50, p52, RelA (p65), RelB, and c-Rel. Under normal conditions, p50 and p52 are produced by the cleavage of p110 and p105 precursors, while RelA, RelB, and c-Rel are not derived from precursor proteins. Typically, NF-κB binds to the inhibitor of nuclear factor kappa-B (IκB), which keeps it in an inactive state. Upon activation of the pathway, IκB Kinase (IKK) phosphorylates IκB, leading to its ubiquitination and degradation. This degradation allows NF-κB to rapidly translocate into the nucleus, where it regulates the expression of numerous genes, particularly those involved in the body’s defense mechanisms, thereby initiating a protective cellular response.

NF-κB signaling can be categorized into classically activated, non-classically activated, and other activated metabolic pathways (Figure 4). In the classical pathway, which is induced by TNF-α, IL-1, or byproducts of bacterial and viral infections, IKK mediates the phosphorylation of Ser32 and Ser36 sites on IκB-α. This degradation of inhibitory proteins allows NF-κB to enter the nucleus and activate gene transcription. The activation of the classical pathway can be divided into two main phases. Initially, an antigen binding to the cell membrane receptor activates phospholipase C, which breaks down membrane phospholipids, producing two secondary messengers: diacylglycerol and inositol trisphosphate. This leads to the stimulation of protein kinase C, which phosphorylates CARMA, Bcl10, and MALT1, thereby activating them. These activated proteins subsequently stimulate the downstream IκB kinase complex, consisting of NEMO, ΙΚΚα, and IKKβ. This complex then phosphorylates IκBα, resulting in its ubiquitination and degradation, which allows p50 and p65 to enter the nucleus and regulate gene expression. The second pro-inflammatory factor, IL-1β, and members of the tumor necrosis factor superfamily—such as TNFα, EDA-A1, and EDA-A2—bind to the tumor necrosis factor receptor. This interaction triggers the recruitment of junction proteins that initiate downstream signaling pathways. The primary molecules that bind to these receptors are from the tumor necrosis factor receptor-interacting factor (TRAF) superfamily, TRAF1-6. Each of these TRAF proteins possesses ubiquitin ligase activity, which facilitates the ubiquitination process and activates the downstream TAK1 kinase. TAK1, in association with its cofactor TAB, phosphorylates the IκB kinase complex, leading to the ubiquitination and degradation of IκBα, freeing p50 and p65. These transcription factors then enter the nucleus to regulate gene transcription, including that of anti-apoptotic and inflammatory genes. Both TNF-α and IκBα participate in corresponding positive and negative feedback regulatory mechanisms (Figure 5).

The non-classical NF-κB pathway is initiated by specific members of the tumor necrosis factor receptor (TNFR) superfamily, such as lymphotoxin-beta or B-cell activating factor. In contrast to the classical pathway, this pathway involves interactions with receptors on the cell membrane that belong to the TNFR superfamily. These interactions primarily lead to the recruitment of TRAF2 and TRAF3 proteins within the TNFR-activating factor superfamily. Intracellularly, NIK interacts with TRAF2 and TRAF3, resulting in their ubiquitination and subsequent degradation.

Upon activation of the non-classical pathway, extracellular ligands bind to their corresponding receptors on the cell membrane, which in turn recruits proteins such as TRAF2 and TRAF3 to the cell membrane’s periphery. This recruitment causes the dissociation of NIK from TRAF2 and TRAF3, leading to an increase in NIK levels. NIK subsequently phosphorylates IKKα, which activates p100. Furthermore, the partially ubiquitinated and degraded p100 is then processed to generate p52, which pairs with RelB to form a heterodimer. This NF-κB heterodimer translocates into the nucleus, where it activates gene transcription that is essential for various biological processes, including lymphocyte homing, B-cell generation, development, and survival, among other immune functions.

In addition to the classical and non-classical pathways, other pathways can also activate NF-κB. For instance, phosphorylation of the Tyr42 site on IκBα occurs in response to various stimuli, such as hypoxia and growth factors. Similarly, phosphorylation of a Ser residue within the PEST structural domain of IκBα by stimuli like ultraviolet light also activates NF-κB signaling. These modifications lead to the ubiquitination and degradation of IκBα, resulting in the release of p50 and p65 dimers. Then, these dimers translocate to the nucleus, where they regulate transcription, particularly the expression of anti-apoptotic genes.

#### 4.2.3. NF-κB Pathway and Major Depressive Disorder

The NF-κB pathway plays a central role in the inflammatory response and immune regulation, processes that are often dysregulated in patients with MDD. Abnormalities in both the inflammatory response and immune system are commonly observed in individuals with MDD, suggesting that the NF-κB pathway may indirectly influence the pathophysiology of MDD by modulating these processes. Leng et al. demonstrated that NF-κB activity was significantly elevated in depression-like mice, and inhibition of NF-κB resulted in alleviation of depression-like behaviors, indicating the pathway’s potential role in MDD [42].

Consequently, the NF-κB signaling pathway has become a target of interest for antidepressant drug development, with several studies focusing on inhibiting NF-κB activity. For instance, Yan et al. found that bergapten reduced depressive symptoms by suppressing both the NF-κB and MAPK pathways [43]. Similarly, melatonin was shown to mitigate depression-like behaviors by inhibiting NF-κB activation [44], while perilla aldehyde exerted antidepressant effects through the downregulation of NF-κB expression [45]. Additionally, targeted therapies have identified specific components of the NF-κB pathway as potential therapeutic targets. Notably, the sigma-1 (σ-1) receptor has been recognized as a promising target for antidepressant treatment. Research indicates that σ-1 receptor deficiency leads to NF-κB pathway activation, which exacerbates depressive behaviors [46].

Inflammation is a defense mechanism that can activate the NF-κB pathway; therefore, anti-inflammatory drugs represent a potential therapeutic strategy for MDD. Glucocorticoids (GCs) are among the most widely used anti-inflammatory agents. GCs exert their effects by binding to the glucocorticoid receptor, which then activates the transcription of anti-inflammatory factors, such as NF-κB, thereby exerting an anti-inflammatory effect [47]. Studies have shown that GR and the NF-κB pathway cooperate to regulate the resolution of inflammation [48]. Furthermore, research on aucubin (AU), the primary component of Eucommia ulmoides Oliv., suggests that AU can improve MDD by promoting GR nuclear expression and inhibiting apoptosis via the NF-κB pathway [49].

Inflammation is often accompanied by apoptosis and pyroptosis, processes that may also be implicated in MDD. Thijssen et al. reported that the NF-κB pathway influences both apoptosis and pyroptosis [50]. Specifically, NF-κB proteins and the genes they regulate have been linked to apoptosis [51]. Certain members of the B-cell lymphoma-2 (Bcl-2) gene family, including Bcl-xL and Bfl-1, as well as other anti-apoptotic genes, can be induced by NF-κB proteins [52]. This supports the idea that the NF-κB pathway is intricately connected to apoptosis regulation, where NF-κB proteins modulate the transcription of both pro-apoptotic and anti-apoptotic genes in response to varying intensities of cellular stimuli (Figure 6) [53].

## 5. Conclusions

The relationship between MDD and stress is complex and multifaceted, making it a critical area of study in mental health research. This paper explored the pathological changes associated with MDD, emphasizing the significant role of stress as a risk factor and the critical involvement of inflammatory responses in its development. The connection between the NF-κB pathway and MDD has been highlighted, offering new insights into the pathway’s role in depression and providing promising directions for future research and clinical interventions. The pathogenesis of MDD remains an ongoing challenge with many unanswered questions. Future studies must aim to broaden sample sizes, account for multiple interfering factors, and refine the understanding of NF-κB pathway targets and related mechanisms. Developing drugs targeting the NF-κB pathway could potentially offer new therapeutic avenues for treating MDD. However, individualized treatments and careful consideration of drug side effects will be crucial in clinical applications. These studies must continue to deepen the understanding of stress-mediated depression pathogenesis, necessitating the advancement of research tools and methods to address these complex issues.

Although current studies acknowledge the importance of stress and inflammation in the pathogenesis of MDD, the findings have not always been consistent, underscoring the complexity of biological, environmental, and psychological factors that contribute to individual variations. Several limitations in existing research have been identified. First, studies have often focused on a single factor, but the prevalence and development of MDD are influenced by a broad range of factors in the clinical context. Second, some research tends to oversimplify the condition of depression, as not all individuals with depression experience suicidal ideation, and not all those who attempt suicide are clinically depressed. Finally, while the NF-κB pathway is recognized as a significant player in MDD, research into its specific targets and related mechanisms remains incomplete.

## Figures and Tables

**Figure 1 cells-14-00629-f001:**
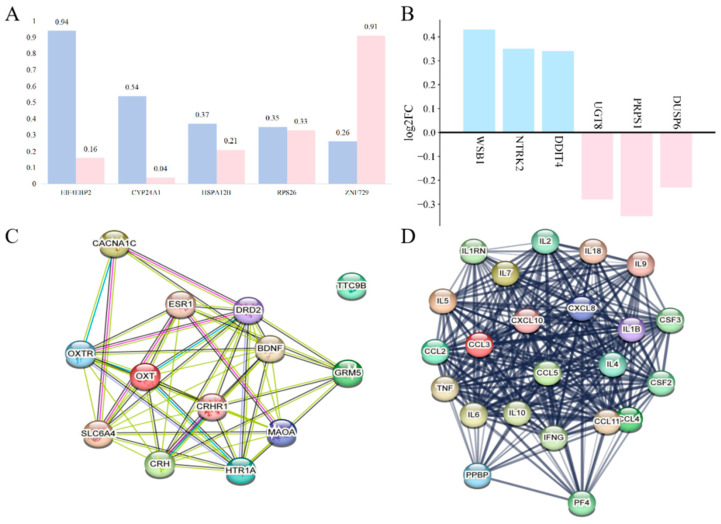
Expression statistics of differential genes and protein interaction network relationships in the brains of MDD models. (**A**): Gene expression in individuals of different sexes with MDD. (**B**): Differential gene expression statistics in the brains of MDD mouse models. The identified genes include: *DDIT4*: DNA damage-inducible transcript 4; *DUSP6*: dual specificity phosphatase 6; *NTRK2*: Neurotrophic receptor tyrosine kinase 2; *PRPS1*: Phosphoribosyl pyrophosphate synthetase 1; *UGT8*: UDP glycosyltransferase 8; *WSB1*: WD repeat and SOCS box containing 1. (**C**): Gene network segment illustrating MDD-related gene expression interactions. (**D**): Protein interaction network of inflammation-related genes implicated in MDD.

**Figure 2 cells-14-00629-f002:**
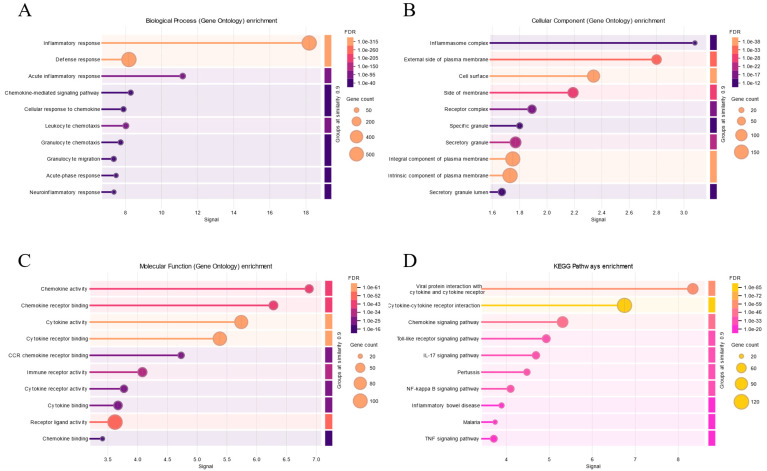
Inflammation-related genes in MDD: gene ontology (GO) and pathway enrichment analysis results. (**A**): In MDD, the function of inflammation related genes is enriched in biological processes by inflammatory response, chemokine-mediated signaling pathway, neuroinflammetory response and etc. (**B**): In MDD, the function of inflammation related genes is enriched in cellular component by inflammasome complex, cell surface and etc. (**C**): In MDD, the function of inflammation related genes is enriched in molecular function by chemokine activity, chemokine receptor binding and etc. (**D**): In MDD, the function of inflammation related genes is enriched in KEGG pathway enrichment by viral protein interaction with cytokine and cytokine receptor and etc.

**Figure 3 cells-14-00629-f003:**
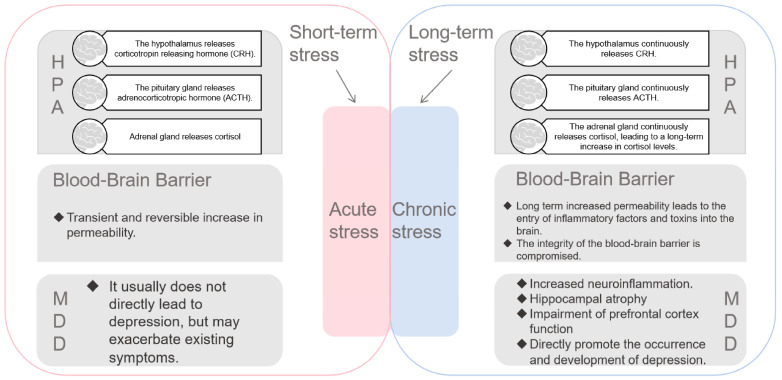
Acute and chronic stress pathways and their effects on depression, hypothalamic pituitary adrenal (HPA) axis, and blood–brain barrier.

**Figure 4 cells-14-00629-f004:**
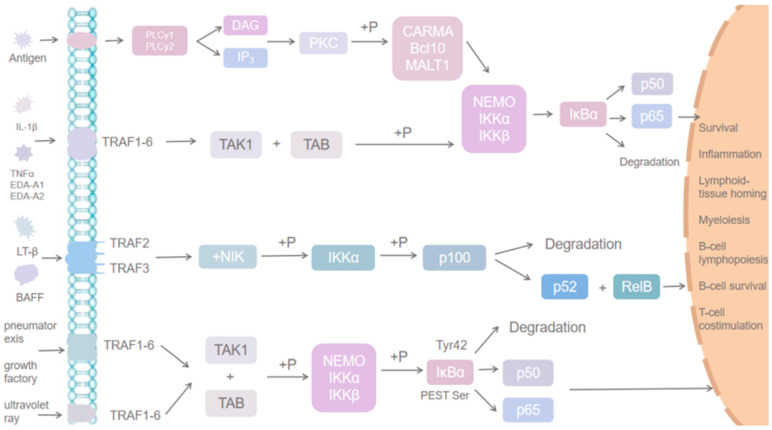
NF-κB activates three metabolic pathways.

**Figure 5 cells-14-00629-f005:**
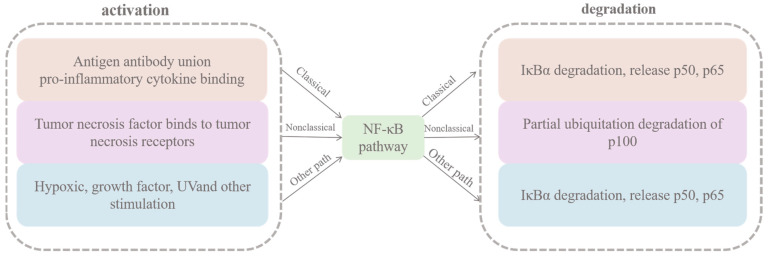
Activation and degradation of NF-Κb.

**Figure 6 cells-14-00629-f006:**
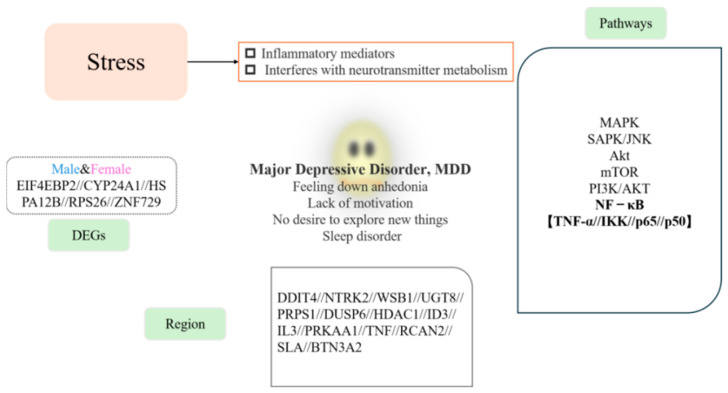
Correlation between stress-mediated inflammation and major depression. This figure illustrates the complex interplay between stress-mediated inflammation and major depression (MDD). DEGs: Represents sex-specific differences in gene expression related to MDD, highlighting variations between males and females. Regions: Denotes specific genetic variations and their associations with MDD, as discussed in the text. Pathways: Represents the biological pathways implicated in MDD, including those involved in stress and inflammation. Center: Depicts the phenotypic manifestations of MDD, such as sleep disturbances, low mood, and other behavioral and psychological symptoms.

**Table 1 cells-14-00629-t001:** Differential expression of inflammation-related genes in patients with MDD.

Symbol	Gene Description	Express	Reference
CCL2	C-C motif chemokine ligand 2	varies	Köhler et al., 2017 [30]; Leighton et al., 2018 [31]
CCL3	C-C motif chemokine ligand 3	up	Syed et al., 2018 [32]; Leighton et al., 2018 [31]
CCL4	C-C motif chemokine ligand 4	down	Syed et al., 2018 [32]; Leighton et al., 2018 [31]
CCL5	C-C motif chemokine ligand 5	up	Syed et al., 2018 [32];
CCL11	C-C motif chemokine ligand 11	up	Leighton et al., 2018 [31]
CXCL4	platelet factor 4	up	Leighton et al., 2018 [31]
CXCL7	pro-platelet basic protein	up	Leighton et al., 2018 [31]
CXCL10	C-X-C motif chemokine ligand 10	Down	Syed et al., 2018 [32];
G-CSF	colony stimulating factor 3	up	Kiraly et al., 2017 [33] Syed et al., 2018 [32];
GM-CSF	colony stimulating factor 2	up	Kiraly et al., 2017 [33]
IFN-γ	interferon gamma	varies	Kiraly et al., 2017 [33]
IL-1β	interleukin 1 beta	up	Kiraly et al., 2017 [33]
IL-1RA	interleukin 1 receptor antagonist	varies	Kiraly et al., 2017 [33]
IL-2	interleukin 2	up	Kiraly et al., 2017 [33]
IL-4	interleukin 4	down	Kiraly et al., 2017 [33]
IL-5	interleukin 5	up	Kiraly et al., 2017 [33]
IL-6	interleukin 6	up	Kiraly et al., 2017 [33]
IL-7	interleukin 7	up	Syed et al., 2018 [32];
IL-8/CXCL8	interleukin 8	varies	Köhler et al., 2017 [30]; Leighton et al., 2018 [31]
IL-9	interleukin 9	up	Syed et al., 2018 [32];
IL-10	interleukin 10	up	Köhler et al., 2017 [30];
IL-12	interleukin 12	up	Köhler et al., 2017 [30];
IL-18	interleukin 18	up	Köhler et al., 2017 [30];
TGF-β1	transforming growth factor beta 1	-	Köhler et al., 2017 [30];
sTNFR2	TNF receptor superfamily member	up	Köhler et al., 2017 [30];
TNF	tumor necrosis factor	up	Köhler et al., 2017 [30];

## Data Availability

No new data were created or analyzed in this study.

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
