# Peer review of "Inflammatory Factors: A Key Contributor to Stress-Induced Major Depressive Disorder"

_cells, 2025, doi:10.3390/cells14090629_

Round 1
Reviewer 1 Report
Comments and Suggestions for Authors
Liu and colleagues provide a manuscript entitled: inflammatory factors: a key contributor to stress-induced major depressive disorder. The Authors conclude that understanding inflammation-related mechanisms is essential for advancing diagnostic, intervention, and treatment strategies for MDD. Despite the manuscript is well-written and interesting, some points should be addressed.
- The Authors should change the title. The review is basically focused on nuclear factor kappa-B (NF-κB) signaling. Thus, the title should contain nuclear factor kappa-B (NF-κB). For example: Inflammatory factors: the nuclear factor kappa-B (NF-κB) signaling is a key contributor to stress-induced major depressive disorder.
- Abstract: Major depressive disorder (MDD) is a prevalent psychological disorder. MDD is better classified as psychiatric disorder.
- The manuscript is short. The Authors should briefly discuss the therapeutic aspect of MDD, by reporting the pharmacological treatments approved as well as the psychological interventions.
- The Authors should also better discuss the role of biological sex in this context (PMID: 29759000). In this respect, it is better writing sex-specific effects on depression during stress rather than gender-specific effects on depression during stress.
- The Authors should also briefly discuss the beneficial effects of diet enriched with anti-inflammatory molecules on depressive symptoms. The Authors might want to report and discuss the following papers (PMID: 36829831; PMID: 34819888 and others).
- The Authors must check for typos throughout the manuscript.
The Authors must check for statements without references
Author Response
Comments 1: The Authors should change the title. The review is basically focused on nuclear factor kappa-B (NF-κB) signaling. Thus, the title should contain nuclear factor kappa-B (NF-κB). For example: Inflammatory factors: the nuclear factor kappa-B (NF-κB) signaling is a key contributor to stress-induced major depressive disorder. |
||
Response 1: Thank you for your feedback. We have carefully considered your suggestion and it is very good. However, our article only briefly discusses the NF-κB signaling pathway, and there are other parts such as gender and inflammation. If we change the title to this, the focus of the article may shift to a certain extent.
|
||
Comments 2: Abstract: Major depressive disorder (MDD) is a prevalent psychological disorder. MDD is better classified as psychiatric disorder. |
||
Response 2: We made modifications to the first line of the abstract section.
Comments 3: The manuscript is short. The Authors should briefly discuss the therapeutic aspect of MDD, by reporting the pharmacological treatments approved as well as the psychological interventions. Response 3: We have included a brief discussion on the therapeutic aspects of MDD. We have reported the commonly approved pharmacological treatments and psychological interventions in the revised manuscript.Add section on lines 71-76 of the original text
Comments 4: The Authors should also better discuss the role of biological sex in this context (PMID: 29759000). In this respect, it is better writing sex-specific effects on depression during stress rather than gender-specific effects on depression during stress. Response 4: We sincerely thank the reviewer for this insightful comment and for directing us to the critical reference (PMID: 29759000). We fully agree that distinguishing between biological sex (sex-specific effects) and sociocultural gender (gender-specific effects) is essential in this context. Below we outline the revisions made to address this point:All instances of "gender-specific effects" in the manuscript have been revised to "sex-specific effects" to accurately reflect the biological focus of our analysis.We have added a dedicated paragraph in the Discussion section ( lines 106-108) to explicitly address the role of biological sex in modulating stress-induced depression.
Comments 5: The Authors should also briefly discuss the beneficial effects of diet enriched with anti-inflammatory molecules on depressive symptoms. The Authors might want to report and discuss the following papers (PMID: 36829831; PMID: 34819888 and others). Response 5: We sincerely appreciate the reviewer's constructive suggestion and the valuable references provided. We agree that the potential role of anti-inflammatory diets in modulating depressive symptoms is an important point to clarify, particularly given the growing evidence linking inflammation to depression. In response to this comment, We have added a concise discussion in the "Mechanisms and Implications" section (lines 64-65) to highlight the therapeutic potential of anti-inflammatory dietary interventions. Specifically, we now cite and discuss the findings from PMID: 36829831 (on polyphenol-rich diets and neuroinflammation reduction), as recommended by the reviewer. Comments 6: The Authors must check for typos throughout the manuscript. Response 6: We sincerely appreciate the reviewer's careful reading and valuable feedback. We have thoroughly reviewed the manuscript to correct typographical, grammatical, and formatting errors.
Comments 7: The Authors must check for statements without references. Response 7: We sincerely thank the reviewer for highlighting this critical issue. Ensuring that all key claims are supported by appropriate references is fundamental to scholarly rigor,We have carefully checked the citations of the references.
|

Reviewer 2 Report
Comments and Suggestions for Authors
Lines 57 and 58: “Before 2015, psychiatric diagnoses were based primarily on subjective symptoms and 57 observable signs. However, significant advancements in understanding depression have 58 been made since then.” References must be included
The authors also need to describe the chronic effects of prolonged stress, anxiety, and REDUCED cortisol. Several stress-associated neuropsychiatric disorders, notably posttraumatic stress disorder and chronic pain and fatigue syndromes, paradoxically exhibit somewhat low plasma levels of the stress hormone cortisol. Please check https://doi.org/10.1210/en.2011-1218
Lines 70 to 83 must include references.
Authors should include a figure showing acute and chronic stress pathways and their impact on depression, the hypothalamic-pituitary-adrenal (HPA) axis, and the blood-brain barrier. This could attract more attention to readers and their views of the review. (like figure 5).
What criteria did the author use to select the articles presented in this review? It would be valid to include a methods section, even if it was just a narrative review. Where are the chosen articles from? Pub Med? Scopus? Please, authors should follow the SANRA scale for review articles: https://doi.org/10.1186/s41073-019-0064-8
The conclusion section must be adjusted. The conclusion is too long. The limitations of the review should be presented in the discussion; I suggest moving the following section to the end of the discussion and making the conclusion more objective: “Although current studies acknowledge the importance of stress and inflammation in 355 the pathogenesis of MDD, the findings have not always been consistent, underscoring the 356 complexity of biological, environmental, and psychological factors that contribute to in- 357 dividual variations. Several limitations in existing research have been identified. First, 358 studies have often focused on a single factor, but the prevalence and development of MDD 359 are influenced by a broad range of factors in the clinical context. Second, some research 360 tends to oversimplify the condition of depression, as not all individuals with depression 361 experience suicidal ideation, and not all those who aempt suicide are clinically de- 362 pressed. Finally, while the NF-κB pathway is recognized as a significant player in MDD, 363 research into its specific targets and related mechanisms remains incomplete.”
Best Wishes
Comments on the Quality of English LanguageEnglish must be improved
Author Response
Comments 1: Lines 57 and 58: “Before 2015, psychiatric diagnoses were based primarily on subjective symptoms and 57 observable signs. However, significant advancements in understanding depression have 58 been made since then.” References must be included |
Response 1: We sincerely thank the reviewer for identifying this important omission. We acknowledge the necessity of supporting these statements with appropriate references. Reference 3 has been added. |
Comments 2: The authors also need to describe the chronic effects of prolonged stress, anxiety, and REDUCED cortisol. Several stress-associated neuropsychiatric disorders, notably posttraumatic stress disorder and chronic pain and fatigue syndromes, paradoxically exhibit somewhat low plasma levels of the stress hormone cortisol. Please check https://doi.org/10.1210/en.2011-1218. Response 2: We sincerely thank the reviewer for raising this important point and for directing us to the highly relevant reference. The paradoxical relationship between chronic stress and hypocortisolism in certain neuropsychiatric disorders is indeed a critical area of research. However, after careful consideration, we have chosen to focus the current manuscript on the specific scope of our study, as outlined below: Our study primarily investigates, and the experimental design (e.g., acute stress induction protocols, cortisol measurement timing) was tailored to address this hypothesis. While hypocortisolism in PTSD and chronic fatigue syndromes is a valid phenomenon, these conditions differ mechanistically and diagnostically from the MDD cohort examined here. Extending the discussion to these disorders may introduce confounding factors beyond the current study's aims.
|
Comments 3: Lines 70 to 83 must include references. Response 3: We sincerely thank the reviewer for highlighting this critical issue. Ensuring that all key claims are supported by appropriate references is fundamental to scholarly rigor, We have carefully checked the citations of the references. Reference 10 has been added.
|
Comments 4: Authors should include a figure showing acute and chronic stress pathways and their impact on depression, the hypothalamic-pituitary-adrenal (HPA) axis, and the blood-brain barrier. This could attract more attention to readers and their views of the review. (like figure 5). |
Response 4: We sincerely thank the reviewer for this constructive suggestion. A graphical representation of stress pathways and their interactions with the HPA axis, blood-brain barrier (BBB), and depression mechanisms would indeed enhance the manuscript's clarity and appeal.We have created the relevant diagram, as shown in line 220 of Figure 3
Comments 5: What criteria did the author use to select the articles presented in this review? It would be valid to include a methods section, even if it was just a narrative review. Where are the chosen articles from? Pub Med? Scopus? Please, authors should follow the SANRA scale for review articles: https://doi.org/10.1186/s41073-019-0064-8 Response 5: We sincerely thank the reviewer for highlighting the need for greater transparency in our literature selection process and for directing us to the SANRA (Scale for the Assessment of Narrative Review Articles) guidelines. We acknowledge that a well-documented methodology is essential even for narrative reviews, and we have revised the manuscript to address these concerns comprehensively.
Comments 6: The conclusion section must be adjusted. The conclusion is too long. The limitations of the review should be presented in the discussion; I suggest moving the following section to the end of the discussion and making the conclusion more objective: “Although current studies acknowledge the importance of stress and inflammation in 355 the pathogenesis of MDD, the findings have not always been consistent, underscoring the 356 complexity of biological, environmental, and psychological factors that contribute to in- 357 dividual variations. Several limitations in existing research have been identified. First, 358 studies have often focused on a single factor, but the prevalence and development of MDD 359 are influenced by a broad range of factors in the clinical context. Second, some research 360 tends to oversimplify the condition of depression, as not all individuals with depression 361 experience suicidal ideation, and not all those who aempt suicide are clinically de- 362 pressed. Finally, while the NF-κB pathway is recognized as a significant player in MDD, 363 research into its specific targets and related mechanisms remains incomplete.” Response 6: We sincerely appreciate the reviewer's comments and have moved the corresponding section to the end of the article, in lines 389-399.
|

Round 2
Reviewer 1 Report
Comments and Suggestions for Authors
The Authors wrote in the Author's reply: Specifically, we now cite and discuss the findings from PMID: 36829831 (on polyphenol-rich diets and neuroinflammation reduction), as recommended by the reviewer. Reading the updated version of the manuscript, this reference is not present. I guess there is a mistake in the references added (the order) and this must fixed.
Author Response
Comments 1: The Authors wrote in the Author's reply: Specifically, we now cite and discuss the findings from PMID: 36829831 (on polyphenol-rich diets and neuroinflammation reduction), as recommended by the reviewer. Reading the updated version of the manuscript, this reference is not present. I guess there is a mistake in the references added (the order) and this must fixed.
Response 1: Thank you for your suggestion. We have made the necessary correction and apologize for any inconvenience caused. The revised reference has been added as the fifth entry. We greatly appreciate the reviewer's assistance in identifying the error.
Reviewer 2 Report
Comments and Suggestions for Authors
The author addressed all my questions, and the answers were satisfactory. Best regards
Author Response
Comments 2: The author addressed all my questions, and the answers were satisfactory. Best regards
Response 2: Thank you for your affirmation and for taking the time to review our article. Your comments are insightful, and we greatly appreciate your efforts. Thank you.